# The Role of the Chronotype in Developing an Excessive Body Weight and Its Complications—A Narrative Review

**DOI:** 10.3390/nu17010080

**Published:** 2024-12-28

**Authors:** Marta Pelczyńska, Małgorzata Moszak, Julita Wojciechowska, Anita Płócienniczak, Jan Potocki, Joanna Blok, Julia Balcerzak, Mikołaj Zblewski, Paweł Bogdański

**Affiliations:** 1Department of Treatment of Obesity, Metabolic Disorders and Clinical Dietetics, Poznań University of Medical Sciences, 49 Przybyszewskiego Street, 60-355 Poznan, Poland; mmoszak@ump.edu.pl (M.M.); pbogdanski@ump.edu.pl (P.B.); 2Faculty of Medicine, Poznan University of Medical Sciences, 70 Bukowska Street, 60-812 Poznan, Poland; 78644@student.ump.edu.pl (J.W.); 88918@student.ump.edu.pl (A.P.); 88910@student.ump.edu.pl (J.P.); 89105@student.ump.edu.pl (J.B.); 89123@student.ump.edu.pl (J.B.); 88824@student.ump.edu.pl (M.Z.)

**Keywords:** chronotype, circadian rhythm, body weight, obesity, metabolic disorders

## Abstract

The chronotype, the personal predisposition towards morning or evening activities, significantly influences health conditions, sleep, and eating regulations. Individuals with evening chronotypes are often at a higher risk for weight gain due to misalignment between their natural tendencies of functioning and social schedules, resulting in insufficient sleep, disruptions in eating habits, and decreased physical activity levels. Often, impaired glucose tolerance and changes in melatonin, adiponectin, and leptin secretion, along with alterations in the clock gene functions in subjects with evening preferences, may be predisposed to obesity. These disturbances contribute to metabolic dysregulation, which may lead to the subsequent onset of obesity complications, such as hypertension, type 2 diabetes, sleep apnea, and liver diseases. Targeting critical components of the circadian system and synchronizing people’s chronotypes with lifestyle conditions could deliver potential strategies for preventing and treating metabolic disorders. Thus, it is recommended to take a personalized chronobiological approach to maintain a normal body weight and metabolic health. Nevertheless, future studies are needed to identify the clear mechanisms between the chronotype and human health. This article provides a narrative review and discussion of recent data to summarize studies on the circadian rhythm in the context of obesity. The manuscript represents a comprehensive overview conducted between August and November 2024 using the National Library of Medicine browser (Medline, Pub-Med, Web of Science).

## 1. Introduction

The chronotype is an individual’s preference concerning biological and behavioral habits dependent on the circadian rhythm dictated by the oscillations of light levels in the environment during the solar cycle [1]. Researchers created several surveys to determine one’s chronotype, including the Morningness-Eveningness Questionnaire (MEQ), which analyzes a subjective approach to performing activities in a 24 h cycle, the Composite Scale of Morningness (CSM), which puts additional focus on shift work, and the Munich Chronotype Questionnaire (MCTQ), which collects sleep and wake patterns on work and work-free days [2]. Despite the methods described above demonstrating high correlations with each other [3], MEQ is considered to be the best tool to evaluate the differences in peoples’ traits according to the circadian rhythm [4]. Depending on the subject’s body temperature and the intensity of their alertness during the day, the chronotype is divided into three groups: morningness chronotype (MC), intermediate chronotype (IC), and eveningness chronotype (EC) [5]. In the last few years, a topic related to the chronotype has become of great interest to researchers.

It is widely known that the circadian clock determines behavior and lifestyle decisions. It impacts not only people but also other mammals; therefore, it may be concluded that it is a primal form of regulating the metabolism, nutrition, and habits affecting the subjects’ physiology [6]. An awareness that chronotype influences dietary patterns, the sleep cycle, and the level of physical activity resulted in research showing that obesity may be associated with the EC. Moreover, the MC seems to be related to more effective weight loss and adherence to healthier patterns, such as the Mediterranean diet [7,8]. An eveningness preference corresponds to the minor allele of the CLOCK 3111T/C polymorphism, resulting in greater difficulties in weight loss [9]. Subjects with the EC are potentially more predisposed to type 2 diabetes mellitus (T2DM) and higher hemoglobin A1c levels (HbA1c) as a consequence [10]. They are also at a higher risk of other metabolic disturbances, as in people with EC, the level of ghrelin and glucagon-like peptide-1 (GLP-1) is changed. These hormones affect appetite, gastric passage, and insulin secretion; therefore, their level of rotation often results in impaired control over postprandial glucose excursions and increased food intake, leading to weight gain [11].

It has been demonstrated that chronotype categories could have an impact on the risk of developing obesity and obesity-related comorbidities. An excessive body weight predisposes one to low-grade inflammation, which correlates with higher cardiovascular risk factors. This is due to various components, including impaired gut absorption resulting in lower iron levels, myocardial overload leading to an anatomical reconstruction of the heart, endothelial alterations that induce hypertension [12], and a higher possibility of developing T2DM. The EC is less likely to maintain a healthy lifestyle. Therefore, people with this chronotype experience numerous complications due to previously mentioned changes. This may result in heart failure, myocardial infarction, coronary artery disease, angina pectoris, and atrial fibrillation [13]. Furthermore, excess adipose tissue is associated with hormonal impairment and cellular damage, causing female infertility, which is why the MC is proven to have a higher chance of reproducing [14]. Obese individuals tend to undergo severe infections and be less responsive to vaccines due to their weakened immune system [15]. Finally, the EC is correlated not only with a greater predisposition to weight gain but also with more difficulties in maintaining weight loss [7].

Given the rising prevalence of overweight and obesity in many countries [16], this review aims to describe a relationship between a chronotype and the development of excessive body weight and its subsequent complications. The manuscript represents a comprehensive overview conducted between August and November 2024 using the National Library of Medicine browser (Medline, PubMed, Web of Science). The following keywords were used: chronotype, circadian rhythm, obesity, eating behavior, physical activity, hypertension, metabolic syndrome, diabetes, sleep apnea, and liver diseases (Appendix A: search terms). Thus, this work provides a narrative review and discussion of recent data to summarize studies on the circadian rhythm in the context of obesity. The structure of the article is as follows. First, the definition and the determinants of the chronotype are described. Secondly, eating behaviors in the context of the chronotype are reviewed. In the next section, the significance of physical activity is revised. In the last section, the relationship between the chronotype and obesity complications is discussed. Finally, the conclusion with a summary of the current state of research is given.

## 2. Determinants of the Chronotype

The chronotype has been defined as an individual characteristic associated with optimal functioning during the day or at night. Although people are classified as a diurnal species due to individual preferences, habits, and predispositions, they can be further divided into three main types of chronotypes, as previously mentioned [17].

The main difference among these groups concerns the sleeping pattern. Subjects with the MC, also called “lark”, prefer to fall asleep and rise early to carry out activities in the first part of the day. On the contrary, people characterized by the EC, called “owls”, usually wake up later and tend to schedule their activities for the late afternoon or evening. While people with the morning chronotype take less time to feel fully alert upon awakening, the evening chronotype reaches its productivity peak in the second half of the day [18] (Figure 1). Nevertheless, most people (about 60%) are classified as the IC, showing characteristics between the morning and evening types. Therefore, they usually present greater flexibility in adapting to daily life without any particular period of time [17]. Moreover, a person identified as the MC or the EC may exhibit its traits to varying degrees (extreme, moderate, or low) [19].

It has been suggested that individual chronotypes also vary in the levels of classic markers for measuring a mammal’s circadian rhythm, including melatonin secretion, the minimum core body temperature, and plasma cortisol levels (Figure 1). Melatonin is a hormone produced by the pineal gland, with its highest concentration observed at night when its secretion is not suppressed by light. Its synthesis is regulated by the suprachiasmatic nucleus (SCN), which serves as the central circadian clock in mammals. Information about the absence of light from the retina reaches the SCN in the hypothalamus. This information is then transmitted via a neural pathway that releases noradrenaline to pinealocytes, stimulating the synthesis and release of melatonin [20]. Endogenous melatonin secretion starts around 2 h before bedtime with a night peak nearly 30 times higher than daytime values. Its production falls during the second half of the night and is inhibited by light. This hormone acts as an internal rhythm synchronizer, promoting sleep by inducing vasodilation and decreasing the core temperature. For this reason, it is considered the most accurate predictor of sleep beginning. It is worth adding that the onset, acrophases, and offset of melatonin secretion occur approximately 3 h earlier in the MC compared with the EC, with no differences in amplitude [19].

It has been shown that, within the rhythmic profile, the onset of melatonin secretion under dim light conditions (dim light melatonin onset, DLMO) is perceived as the most valuable marker for assessing the circadian pacemaker. Similarly to melatonin secretion, the MC presents earlier higher DLMO values, while the EC exhibits an increase in DLMO values later [21]. Zerbini et al. showed that the season and weekly structure are related to changes in the phase of entrainment and sleep. During the summer, sleep times and the DLMO were about 1 h earlier compared to winter, while sleep duration was shorter. Moreover, the DLMO and sleep times were later on work-free days, and their phase differed in workdays. What is important is that these effects were stronger in the EC [22]. In accordance with these findings, in a meta-analysis from the year 2020, Wei et al. indicated that the melatonin level in night-shift workers was significantly lower than in day workers (SMD = −0.101, 95% CI = −0.179 to −0.022, *p* = 0.012), which suggests the suppression of melatonin production in this population group [23].

As previously mentioned, people with various chronotypes differ according to body temperature. The results from studies involving the measurement of minimum body temperature evaluated a 2–3-h difference in the positions of the circadian phase between the MC and EC [24,25,26]. The comparison of the Tmin between genders emphasized that it typically occurs roughly thirty minutes later in men than in women. Additionally, temperature amplitudes calculated only in male participants were larger in subjects with more delayed phases due to lower nocturnal temperatures [24].

Cortisol, often described as the “stress hormone”, is a steroid compound produced by the adrenal glands, and its secretion is regulated by the hypothalamic–pituitary–adrenal (HPA) axis. It is needed for controlling various functions, such as the metabolism of glucose, proteins, and lipids, as well as managing blood pressure and modulating the immune response [27]. Chronic stressful situations can predispose one to an imbalance in its homeostasis [28]. Cortisol levels show distinct diurnal variations related to the sleep–wake cycle, peaking shortly after waking up and gradually decreasing to their lowest levels around midnight [29]. Bailey et al. conducted a study involving 19 healthy adults, with 10 identifying as the MC and 9 as the EC. Blood samples were collected every 2 h over 38 h and analyzed for serum cortisol concentrations. The results showed that the cortisol acrophases occurred 55 min earlier in the morningness group and that the eveningness group had a smaller cortisol rhythm amplitude [30]. Moreover, Randler and Schaal analyzed saliva samples from 43 adolescents and 82 young adults upon awakening and 30 min later. Their findings indicated that individuals with the MC exhibited higher cortisol levels immediately after waking up [31]. It seems that the more morning-oriented chronotypes, the higher the level of cortisol secretion is observed [32]. These findings may explain why people with the MC achieve full wakefulness more quickly and can undertake various activities effectively in the morning compared to the EC.

A person’s sleep preferences appear to be determined by environmental and genetic factors. It is estimated that the latter may contribute up to 50% of the variability observed in chronotypes. Several genes, known as circadian clock genes, have been identified as potential candidates for their association with various chronotypes [33].

Clock genes play a crucial role in regulating the circadian rhythm of living organisms through a circadian feedback loop. This loop starts with the expression products of the *CLOCK* (*Circadian Locomotor Output Cycles Kaput Gene*) and *BMAL1* (*Brain and Muscle ARNT-like 1*) genes forming a heterodimer BHLH–PAS (Basic Helix-Loop-Helix-Per-Arnt-sim), via their PAS domain and bind to the E-box through the BHLH domain, which initiates the transcription of other genes, such as *PER* (*Period gene*) and *CRY* (*Cryptochrome*). These proteins accumulate in the cytoplasm, and after reaching an ultimate concentration, they form a complex that translocates to the nucleus. Within the nucleus, they inhibit the activity of CLOCK–BMAL1, consequently suppressing their transcription. The cycle begins anew once *PER* and *CRY* are degraded [34]. This time-delay cycle through transcription and subsequently the accumulation/degradation of proteins modifies a negative-feedback homeostat into a self-sustained oscillator (around 24 h) [35]. Other regulatory genes, such as *RORs* (*Retinoic acid Receptor–related Orphan Receptors*) and the REV-ERB proteins *NR1D1* and *NR1D2* (*Nuclear Receptor subfamily 1, group D, member 1 or 2*), respectively, activate or inhibit the transcription of BMAL1 by competing in binding to the mentioned gene [36].

Several studies evaluated a candidate gene approach to assess the relationship between circadian genes and the chronotype. Results showed that individuals carrying one of the two *CLOCK* alleles, 3111C, exhibited a notably lower average Horne–Ostberg score, indicating a preference for eveningness [37,38]. In other studies, morningness has been found to be associated with polymorphisms in the clock genes *PER1* and *PER2* [39,40]. A relationship was also noticed between a length polymorphism in the *PER3* gene and the occurrence of a specific type of chronotype. Archer et al. compared the variable number of tandem repeats (VNTR) in the *PER3* gene among individuals from two groups with extreme scores on the MEQ scale (approximately 7% of 484 volunteers), as well as a control group of equal size consisting of individuals with intermediate scores. It was found that individuals in the EC group more frequently exhibited four repeats than those in the MC group, where five repeats were more common [41]. Although other researchers confirmed these results [42,43], it should be mentioned that there are studies in which no significant correlation between these specific gene variants and types of chronotypes occurred [44,45]. The obtained statistical data exhibit features of a normal distribution, which suggests that there is not a single gene responsible for determining a person’s chronotype, but it is rather the result of an additive, polygenic effect [33]. For this reason, a more current approach is to perform a genome-wide analysis to identify new genetic loci, as in the study by Jones et al. The researchers conducted the most extensive Genome-Wide Association Study (GWAS) on the self-reported chronotype, involving 697,828 participants. The authors identified 351 loci linked to the chronotype preference, of which 24 had been previously documented in an earlier GWAS, and 327 were newly discovered, i.e., *PER1, PER2, PER3, CRY, FBXL3,* and *ARNTL* [46]. Similar results were obtained earlier in other GWASs [47,48]. What is more, Molina-Montes et al. established that genetic variants of some circadian clock genes (variant rs2735611 of the *PER1* gene and three *CLOCK* gene variants—rs12649507, rs3749474, rs4864548) could explain the link between genetic susceptibility to the obesity risk and an individual’s chronotype [49].

The chronotype is a relatively constant feature of a given individual, but different age groups display specific tendencies. Typically, young children lean towards an MC before entering their teenage years. Conversely, in the group of adolescents and young adults, an EC is usually observed [2]. This becomes especially visible among college students transitioning to independent living [50]. Then, over time, people stand out to demonstrate earlier sleeping preferences, and the elderly usually show a renewed tendency to be morning-oriented [51,52].

Gender appears to be a factor influencing a person’s chronotype as well. Most studies have observed a more frequent occurrence of the MC in females compared to the EC in males. This may be due to hormonal differences and societal gender roles [53]. Exposure to light also seems to significantly impact the type of chronotype. It has been proven that people born during a season with reduced sunlight intensity, i.e., during a short photoperiod (autumn, winter), have a greater tendency to show a morning chronotype than those born during a long photoperiod, indirectly confirming an imprinting-like phenomenon conditioned by the photoperiod at birth [54]. Additionally, those who classify themselves as the “MC” tend to have more natural light exposure in the morning, whereas the “EC” is exposed to more artificial light during the evenings and at night [55]. Moreover, after four hours of exposure to blue light, results from an electroencephalogram (EEG) indicated that subjects with the MC had elevated theta and low alpha spectral power in the afternoon in comparison to people exposed to polychromatic white light, suggesting that MC individuals may be more sensitive to light [56].

Eventually, social factors may impact the chronotype as they often interfere with subjects’ sleeping preferences. Since exogenous factors, such as the school schedule or shift work, exist, there is often a misalignment between sleep time and activity between life rhythmicity and the biological clock. This phenomenon has been called “social jetlag” [57]. Evening types are particularly vulnerable to social jetlag, as most activities require people to start their efforts early. Therefore, it is unsurprising that people with later tendencies are likelier to report poorer sleep quality and daytime sleepiness [58]. Conversely, research on nurses working night shifts showed that those with an earlier chronotype had more difficulty adapting to their work schedule and suffered more significant social jetlag [59]. In the short term, this misalignment between social demands and the chronotype may decrease productivity. Still, in the long term, it may hurt overall health and potentially lead to various diseases [60,61].

Ekiz Erim and Sert described an interesting finding. The authors investigated the effect of the Circadian Timing Program (SİZAP) developed for EC subjects with the first degree of obesity on obesity management and sleep quality. SİZAP includes instructions for regulating daylight exposure, sleep, caffeine and food intake, and exercise times to ensure proper hygiene training. The research group had a statistically significant decrease in daytime sleepiness and the anthropometric measurements (*p* < 0.05). Moreover, the scores of the bodily functions sub-dimension of the quality of life scale and the sleep quality scores were better in the SIZAP group (*p* < 0.05), showing that this therapy may be an effective target in obesity treatment [62]. Muscogiuri also indicates that the physiological timing of energy intake may have anti-obesity potential [63].

## 3. Chronotype and Eating Behavior

The chronotype influences the time meals are eaten and determines preferences, as well as eating habits. It regulates the intake of specific food groups and the total energy per day. Most people distribute their meals according to their biological rhythm rather than the societal schedule. Morning chronotypes consume more energy early in the day, whereas EC shifts their meals to later hours, closer to the evening [64]. Such a difference in meal timing causes the EC to present a greater tendency to omit breakfast [65,66]. A study involving 1854 adults aged 25–74 years found that the EC, unlike morning chronotypes, eats 4–5% less energy in the morning and, at the same time, up to 6–7% more energy in the evening hours [67]. Also, another study conducted by Lucassen et al. on 114 obese individuals showed higher calorie consumption in the mentioned chronotype. It was found that individuals with an EC exhibit approximately 50% greater calorie intake after 8:00 p.m. compared to those with an earlier chronotype [68]. Skipping breakfast and shifting meals to later parts of the day may have negative health implications. Research shows that such dietary patterns affect the cardiometabolic profile [69]. Skipping breakfast may lead to weight gain, especially in individuals who prefer the evening time. The NHANES study on individuals aged 20–39 demonstrated that breakfast consumption reduces the risk of obesity and overweight [70]. Conversely, a study on men from the United States aged 46–81 found that eating breakfast reduces the risk of gaining weight (up to 23%) [71]. Consuming energy after 8:00 p.m. may increase the body mass index (BMI) [72]. Studies also indicate that late dinner consumption is associated with elevated levels of triglycerides and cholesterol [73]. Moreover, consuming large portions of meals during dinner is linked to poorer glycemic control [74].

Individuals who prefer the evening time exhibit additional unhealthy eating habits, as presented by Teixeira et al. in their systematic reviews [75]. They tend to eat more significant portions and opt for a second one. Additionally, they demonstrate a heightened tendency towards emotional eating [76] and eat less frequently than MC [68]. It has been observed that people with the EC have more irregular meal patterns [67]. In a systemic review and meta-analysis conducted by Rodríguez-Cortés et al., a higher tendency of incorrect eating behaviors was observed in children and adolescents with the EC suffering from overweight/obesity [77]. Research indicates that such a situation may contribute to the development of insulin resistance (IR) or increased risk of metabolic syndrome (MetS) [78].

The evening chronotype type consumes more caffeine and alcohol [79]. The higher consumption of these beverages is likely associated with their stimulating properties [80]. A study on 2854 Thai college students revealed that individuals, particularly those with an EC, who frequently experience sleep deficits, consumed higher amounts of caffeine to improve focus and concentration while studying [81]. Moreover, Bodur et al. have pointed out a correlation between poorer sleep quality in the EC and increased caffeine consumption in coping with social obligations [82]. A meta-analysis of 13 studies showed that EC individuals were 41% more likely to consume alcohol versus other chronotypes (OR = 1.41, 95%; CI = 1.16–1.66; *I*^2^ =  38.0%) [83].

However, no exact conclusion exists regarding differences in daily energy intake between the MC and EC. Some studies suggest that daily energy intake is the same regardless of the chronotype [67,84]. Others show higher energy intake among ECs, as previously mentioned [85,86,87]. There is also inconsistency regarding the types of foods consumed by different chronotypes. Some studies show no differences in macronutrient intake between chronotypes [66,68]. However, a survey conducted by Toktas et al. on 23 male university students found higher carbohydrate and fat consumption and lower protein intake in the EC [85]. Similarly, a study by Sato-Mito on 3304 Japanese female students aged 18–20 indicated lower protein intake in the evening type [88]. Another study involving 4493 adult participants confirmed higher fat and lower protein intake in this chronotype. However, it was found that there was a lower consumption of carbohydrates in the EC [89]. This is also confirmed by a study on 223 German teenagers, where lower carbohydrate intake was also observed in this chronotype [90]. Conversely, a study conducted by Arslan et al. on 204 healthcare workers showed higher carbohydrate intake among individuals who prefer an evening time [86].

Another difference between chronotypes occurred in vegetable and fruit consumption. In a study on 511 British teenagers, the EC was found to have a higher frequency of consuming unhealthy snacks and a lower intake of vegetables and fruits throughout the day [91]. Similar results were obtained in another study by Baron et al. on 52 participants, which showed that individuals with an EC consume fewer vegetables and fruits, as well as more carbonated beverages and fast foods [72]. However, one scoping review shows that while vegetable consumption is lower in the EC, fruit consumption seems similar in both chronotypes [64].

It appears significant that sleep quality, which is observed to be poorer in the EC, influences dietary choices and total energy intake throughout the day [92]. A study of 3072 Istanbul/Turkey residents found that individuals with poorer sleep quality consumed more protein, fiber, and cholesterol, while those with good sleep quality consumed less total energy and carbohydrates. Additionally, men exhibited lower fat intake [92]. A study carried out by Weiss et al. on 240 teenagers also revealed a link between a shorter sleep duration and higher fat consumption. It was found that individuals who slept less than 8 h consumed more energy from fat food than those who slept more [93]. Furthermore, a meta-analysis identified a relationship between higher energy and macronutrient intake and lower sleep quality [94]. These results indicate that poor sleep quality, associated with the EC, leads to increased energy and macronutrient consumption, especially fats.

As observed, the EC tends to shift their meals to later parts of the day, resulting in lower energy intake in the morning and higher intake in the evening. Additionally, they exhibit a higher frequency of skipping breakfast, irregular meal patterns, and unhealthy eating behaviors. They are also characterized by higher consumption of alcohol and caffeine. However, research is inconclusive regarding overall energy and macronutrient intake throughout the day, but the quality of sleep appears to have a significant impact on these aspects.

## 4. Chronotype and Physical Activity

The evening chronotype seems to be related to many behaviors that impair human health. This relationship is also noticeable in the context of physical activity (PA) [87,95]. The association between the chronotype and PA level is especially noticeable in the population around the mid-twenties [96]. A low level of PA, which is associated with eveningness, is one of the factors leading to health problems and disorders, including gaining weight and being overweight. PA alleviates the negative results on well-being caused by eveningness [97], while an insufficient level of PA is associated with higher depressive symptoms [98,99,100].

There are many physiological factors influencing physical performance that have daily fluctuations. The ones of significant importance are cortisol and body temperature. Morning cortisol levels stimulate metabolic pathways, leading to skeletal protein turnover. Intense physical exercise raises cortisol, which is associated with decreased sports performance due to its catabolic effect and inhibiting protein synthesis [101,102]. The body temperature increases in the early evening [103]. It positively impacts muscle tissue by modifying components of muscular contractions, like calcium uptake and the activity of myosin ATPase [101]. Thus, what is interesting, considering the influence and daily fluctuations in cortisol and body temperature, is that it can assumed that physical activity undertaken in the second part of the day may be more efficient [102]. Additionally, it has been proven that the work rate of exercises shorter than 50 min is higher in the evening [103].

The chronotype seems to influence the total level of PA during the day [87]. As previously mentioned, individuals with an MC tend to be more physically active than people with an EC [95,96]. This correlation was demonstrated between the chronotype and level of physical activity in all its categories: light, moderate, and muscle strengthening [104]. A study involving 4904 adults aged 25–74 years found that the percentage of people with an EC whose PA level is none or very small (28%) is more than twice higher than in the MC group (12.2%) [99]. The amount of time spent on PA and the frequency of PA also differ between chronotypes. A study on 202 healthy women aged 21–65 confirmed that the MC correlates with a higher frequency of PA. The level of PA in the evening-type group was 2.28 (SD = 1.5) times per week, while in the morning-type group, the score was 2.85 (SD = 1.65) times per week (*p* = 0.042) [100]. This relationship is not only related to the impartial PA level but also to a worse subjective fitness level among ECs [105]. Thus, a low level of PA impacts sedentary behavior, especially in people who represent an EC [96]. Considering all sedentary activities, including those related to work, television viewing, computer use, vehicular travel, and other contexts, it has been observed that individuals with an EC exhibit a higher sitting index. The study by Wennman et al. showed that 20.5% of individuals with an MC demonstrate a high sitting index, while the score in the EC group for this category was nearly two times higher at 35.9%, Table 1 [99].

In 2020, Merikanto et al. published the results of a study in which they demonstrated that adolescents exhibit a greater tendency towards eveningness, and this shift is reflected by a decline in PA levels, which progressively decrease from early to late adolescence. This reduction in PA participation throughout adolescence may increase the risk of obesity and cardiovascular issues in adulthood [105]. Moreover, in a recent meta-analysis from 2024 conducted by Huang et al., it was shown that the MC is associated with a higher PA level and a lower level of sedentary behavior than the EC [106].

Not only is the amount of time spent on PA diverse in groups of different chronotypes, but the physical performance also has different scores. During the comparison of morning race times, individuals with an MC demonstrate superior athletic performance compared to people with an IC and EC [102]. However, Brown et al. showed that when morning and evening scores of the same group are compared, the difference in scores is more significant for the MC group than for the group of people with an EC or IC [107].

Based on a study conducted by Rae et al. in 2015, most MC-oriented people score better in the morning, and most EC-oriented people score better in the evening. However, these individuals who do not have better scores in the part of the day, dependent on their chronotype, swipe training hours during the day. For example, MCs who score better in the evening have trained in the late hours. It may be concluded that the training time affects the hours of the best scoring despite the chronotype [108]. Another element influencing physical performance is the elapsed time after waking up. Facer-Childs and Brandstaetterin indicated that individuals with the EC have the best physical performance at 11.18 ± 0.93 h after waking up, with people with the IC after 6.54 ± 0.74 h and MC-oriented after 5.60 ± 1.44 h. This means that individuals with the EC need twice as long as those with the MC to be active enough to score at the same performance level, Table 1 [109].

Numerous components of physical performance vary across different chronotypes (Figure 2). Most of these variations suggest that individuals with a more pronounced EC tend to exhibit poorer athletic performance. The first one is the change in power. A study conducted by Lim et al. on 340 elite athletes found that the power drop (%) is bigger in individuals with an EC than with an MC. At the same time, the mean power (measured both in W and in W/kg) and peak power (also both in W and W/kg) in a group of people with the MC are significantly higher than in a group of individuals with the EC [110]. The second is oxygen consumption. The data from the study indicated by Hill et al. indicated that EC-oriented people have higher maximum oxygen consumption in the evening than in the morning. However, this difference is not noticed in the MC group [111]. Another factor is the heart rate recovery time (HR) after PA. Sugawara et al. showed that people with the EC need more time to normalize their HR in the morning than those with the MC at the same time of the day or EC-oriented people in the evening. This may result from slower post-exercise reactivation of the vagus nerve for individuals with the EC in the early morning [112]. When analyzing the subjective rating of perceived exertion (RPE), there has not been any diurnal variation in the RPE, even despite a higher evening work rate [103]. Nevertheless, dissimilarities in the RPE were noted in a study conducted by Rae et al. on a group of 26 swimmers based on the timing of the individuals’ training sessions. Those who engage in morning training experience reduced perceived effort during morning exercise sessions, whereas individuals training in the evening exhibit a lower RPE during evening workouts. This is indirectly related to the chronotype as it affects the time of training. This disparity was noticed only after the warm-up, and no RPE change was detected after the whole training session [108]. A study on 20 cyclists revealed that when examining the direct correlation between the chronotype and RPE, it is noticeable that people with the MC experience a lower RPE after self-paced or submaximal physical activity in the morning, compared to their RPE levels in the evening, despite maintaining the same relative intensity workload. However, this relationship is not observed following maximum-intensity performance [113].

Levels of fatigue and vigor may have an impact on physical performance. Individuals with the MC are less fatigued than people with other types of chronotypes in the first part of the day [102]. Thus, MC-oriented people have lower fatigue and higher vigor in the morning than in the evening (Figure 2). In contrast, people with the IC and EC show the opposite pattern—higher fatigue and lower vigor in the morning compared to the evening, Table 1 [108]. In the already mentioned study, Wennman et al. also showed that individuals with the MC have more liveliness in the morning, while those with the EC have much higher alertness in the afternoon or evening. When comparing morning alertness in MC and EC groups, 49% of the MC describe it as good, while 70% of the EC define it as difficult, which may be the reason for the differences in levels of fatigue and vigor noted before [99].

Not only do the physical aspects of PA vary according to the chronotype, but the chronotype also influences specific psychological characteristics, such as physical activity self-efficacy, and the positive attribution of PA effects. A study involving 726 students from German schools aged 10–17 years noticed that individuals with the MC exhibit higher self-efficacy and a more positive assessment of PA’s effects than people with the EC. As a consequence, it seems that individuals with an EC are more predisposed to discontinuing PA engagement, as they do not perceive the outcomes of their PA efforts as positively as individuals with an MC [95].

There has been a noticeable relationship between the chronotype and the choice of sports disciplines. Based on the human body’s response, sports disciplines can be grouped into skill, power, and endurance. There is also a category of mixed sports, including a combination of at least two groups mentioned before [114,115]. A study on 174 African University game players, with a mean age of 21.65 ± 2.05 y, found that IC players mainly participate in skill, concentration, and mixed sports events, excluding football and athletics, which are more likely to be joined by players with the MC [116]. Moreover, MC-oriented people practice endurance, games, and coordinative sports more often [95].

Circadian rhythms may increase people’s levels of PA. Sempere-Rubio et al. concluded that people with stable circadian rhythms have higher PA levels than those with unstable ones. Thus, despite the type of chronotype, keeping a stable circadian rhythm may increase the level of PA, reducing the risk of being overweight [96]. In conclusion with these remarks, in the CORDIOPREV study, it has been observed that subjects with the EC were less active than the MC (201 vs. 251 min/week; *p* = 0.01) and represented a more sedentary lifestyle (750 vs. 659 min/week; *p* < 0.01) (Table 1), which correlated with a higher prevalence of MetS in this population group (OR 1.58 IC 95% [1.10–2.28], *p* = 0.01) [117].

Based on the information provided above, conclusions can be drawn that the chronotype has a meaningful impact on people, also in the context of physical activity. Chronotype preferences should be considered while designing training sessions in academic and professional conditions [96,116]. On the other hand, not all studies [82,118] indicated a clear relationship between the EC and lower levels of PA; thus, a large sample size and more rigorous RCT should be designed for specific populations to fully understand exercise’s effects on different time points.
nutrients-17-00080-t001_Table 1Table 1The characteristics of physical activity according to chronotype.Characteristics of Physical PerformanceIndividuals with the MCIndividuals with the ECResultsStudyElapsed time after waking up needed to score the best physical performance5.60 ± 1.44 h11.18 ± 0.93 hEC needs twice more time after waking up to score the bestFacer-Childs et al. [109]Frequency of PA [times per week]2.85 (SD 1.65)2.28 (SD 1.5)MC has higher PA frequency throughout the weekHaraszti et al. [100]MVPA [min per day]22.3 (17.1 to 27.5)12.5 (7.4 to 17.7) EC displayed the lowest levels of MVPAHenson et al. [119]Meeting the PA guidelines48.2%35.9%EC was less likely to meet physical activity guidelinesMakarem et al. [61]Duration of PA[min per week]357.5 (SD 393.9)285.8 (SD 349.0)MC reported more min of physical activity per week Patterson et al. [120]Level of fatigue in the first part of the day compared to eveningLowerHigherMCs are getting more fatigued as the day goes by, opposite to ECsRae et al. [108]Sedentarism [min per week]695 min750 minECs were less active and more sedentary than MCsRomero-Cabrera et al. [117]Time of better scoresMorningEveningMCs perceived less effort when performing a submaximal physical task in the morning than ECsVitale et al. [102]Sedentary behavior [high sitting index]20.5%35.9%Almost twice more of ECs have a high sitting indexWennman et al. [99]Morning alertnessGood for 49%Difficult for 70%MC has much better morning alertnessPreferred time for a physical task lasting around 2 h8–10 am (chosen by 82% MC)3–5 pm (chosen by 56% EC)Individuals prefer to exercise in part of the day, privileged by their chronotypeAbbreviations: EC—evening chronotype; MC—morning chronotype; MVPA—moderate-to-vigorous physical activity; PA—physical activity; SD—standard deviation.


## 5. Chronotype and Weight Gain

Obesity is a complex health condition in which a person’s chronotype plays a significant role in its development [9]. Individuals with an EC may be especially predisposed to metabolic disorders and irregularities in sleep patterns, which can contribute to weight gain [91]. Those subjects have been reported to tend to follow unhealthy lifestyles mainly characterized by sedentary behavior and the high intake of unhealthy food.

As previously mentioned, clock genes, such as *CLOCK*, *BMAL1*, *PER*, and *CRY*, play a key role in regulating the circadian rhythm. The expression of these genes oscillates in a roughly 24 h cycle, which forms the basis of the biorhythm. The protein products of these genes create complex feedback loops that drive the biological rhythm. The expression of these genes involved in lipogenesis and lipolysis influences adipose tissue deposition and breakdown. Disruptions in the circadian rhythm can promote excessive adipose tissue deposition and contribute to the development of obesity [49,121]. One systematic review showed that the rates of the minor allele (C) genes, associated with obesity, and *SIRT1-CLOCK* genes, related to the resistance against weight loss, are higher in individuals with EC [122].

Insulin, a key hormone responsible for regulating blood glucose levels, exhibits daily fluctuations in its secretion. Individuals with an EC may have poorer insulin sensitivity, especially if they consume meals later in the day [123]. When natural insulin sensitivity is lower, that situation can lead to higher post-meal glucose levels and an increased risk of insulin resistance and T2D [124]. An irregular sleep pattern often accompanies individuals with an EC and can further disrupt the normal daily rhythm of insulin secretion [125]. A lack of adequate sleep or sleeping at inappropriate times can lead to disturbances in insulin secretion and increase the metabolic risk [126]. Research shows that there can be a natural increase in insulin sensitivity throughout the day, which is particularly noticeable in the morning [127]. Individuals with an MC often better utilize this increase in insulin sensitivity by consuming breakfast, which can lead to more stable glucose levels throughout the day [123]. Thus, EC individuals have been associated with a higher Homeostasis Model Assessment of Insulin Resistance (HOMA-IR) values, often due to less adherence to a healthy diet and more unhealthy behaviors and eating patterns **[122]**. What is more, de Almeida et al., in a meta-analysis of randomized crossover studies, showed that the intake of carbohydrates at night leads to higher glycemic (SMD = 1.30; 95% CI, 1.01 to 1.59; *I*^2^ = 0%; *p* < 0.00001), but not insulinemic (SMD = 0.19; 95% CI, −0.10 to 0.49), postprandial values in the evening compared to the morning chronotype [128].

Leptin and ghrelin are hormones that regulate appetite. Leptin signals satiety, while ghrelin stimulates appetite. Their secretion can vary depending on one’s chronotype, affecting dietary preferences and meal timing. Individuals with an EC may have higher levels of ghrelin in the evening, leading to increased appetite and later meal consumption, which can contribute to weight gain [11,129].

Cortisol dysregulation can affect glucose metabolism and increase the risk of metabolic disorders, such as insulin resistance and T2D. Additionally, disrupted cortisol rhythms can lead to increased inflammation and changes in blood pressure regulation, further complicating overall health [130]. High cortisol levels can promote the accumulation of adipose tissue, especially around the abdomen, which is associated with a greater risk of developing cardiovascular diseases and T2D [131].

An irregular sleep pattern can reduce the resting metabolic rate, meaning that the body burns fewer calories while at rest. Research has shown that chronic sleep deprivation can affect the hormonal regulation of metabolism, including hormones, such as cortisol, which can disrupt the normal energy balance [132]. Reduced thermogenesis due to sleep disturbances can also stem from poorer body temperature regulation, which is linked to a decreased ability to burn adipose tissue more efficiently [133]. In agreement with these observations, in one prospective study, it has been shown that the EC predicts more significant weight gain and higher BMI values [134].

A person’s chronotype significantly impacts the circadian rhythm and various metabolic aspects, which can contribute to obesity. The impaired insulin sensitivity, adiponectin, melatonin, clock gene functions, and increased leptin secretion detected in subjects with the EC represent favorable traits for weight gain [129]. Understanding these relationships is crucial in creating effective strategies for preventing and managing overweight and in the broader context of promoting metabolic health [135].

## 6. Chronotypes and the Complications of Obesity

Among the most common complications of obesity in the global population are cardiovascular complications (hypertension, coronary heart disease, heart attack, heart failure, arrhythmias, venous thromboembolic disease, stroke), metabolic complications (metabolic syndrome, prediabetes, type 2 diabetes, atherogenic dyslipidemia, non-alcoholic fatty liver disease, gallstone disease), respiratory system disorders (bronchial asthma, chronic obstructive pulmonary disease, obstructive sleep apnea—OSA), neoplasms (esophageal, colon, rectal, prostate, liver, pancreas cancer), mechanical complications (degenerative joint and spine disease), hormonal disorders (polycystic ovary syndrome), dermatological changes (acne), and urinary system complications (chronic kidney disease, obesity-related glomerulopathies) [129,136,137,138,139,140].

While the complications of obesity mentioned above are well documented in the literature, attempts to link excessive body weight and its complications in the context of the chronotype have been relatively recent and have varying results [141,142]. Due to many obesity-related complications, only selected ones have been discussed in the context of the chronotype in this article.

### 6.1. Hypertension

Hypertension is a significant risk factor for cardiovascular disease and its complications. A relationship between hypertension and obesity is well described in the literature [143]. In humans, blood pressure exhibits a circadian rhythm with a physiological decrease at night [144]. Thus, its relationship with the type of chronotype needs to be established.

Lv et al. conducted a study to investigate the association between sleep quality and the development of hypertension in 165,493 members of the British population. Five sleep-related factors (chronotype, sleep duration, episodes of insomnia, episodes of snoring, and episodes of excessive daytime sleepiness) were used to create a sleep quality score (SQC) from 0 to 5 (where each factor was scored from 0 to 1). A higher score indicated a healthier sleep pattern. The chronotype was determined by a four-answer questionnaire: “Definitely a ‘morning’ person”, “More a ‘morning’ than ‘evening’ person”, “More an ‘evening’ than a ‘morning’ person” and “Definitely an ‘evening’ person”. The “Definitely a ‘morning’ person”, and “More a ‘morning’ than ‘evening’ person” chronotypes were classified as one of the factors for healthy sleep, in contrast to the evening chronotype. Notably, among patients with an SQC score of 5 points, 100% of the participants reported being an MC, while among those with a score of 0–1 points, only 6.1% reported being an MC. All subjects were given the SQC test, BMI was calculated, and BP was measured, both systolic (SBP) and diastolic blood pressure (DBP). The results showed that a measure of worsening sleep quality, including the predominance of the evening chronotype over the morning chronotype, was associated with higher values of both SBP and DBP and a general tendency towards an increased risk of developing hypertension, Table 2 [145].

Another example of a complication of obesity in the context of chronotype is a study by Klawe et al. on a group of 50 male shift workers who analyzed the circadian course of the mean arterial pressure (MAP) after day and night shifts. The subjects were divided into subgroups based on age, the biological chronotype (Horne–Ostberg test, modified by Kwarecki, Zużewicz), obesity, and heart rate-to-respiratory rate ratio (HR/RF), which reflects the imbalance of circadian rhythms. In the subgroup of workers aged 40–55, with obesity and a morning chronotype and HR/RF ratio > 4 (indicating a disturbance in the balance of circadian rhythms), MAP values were significantly higher after the night shift than in other subgroups. The study, despite the small sample size, confirms that the chronotype may play a role in the development of hypertension in obese individuals. The authors also highlight the link between shift work that is not in line with an individual’s chronotype and an increased risk of obesity complications compared to those whose working hours are matched to their chronotype, Table 2 [146]. On the other hand, other studies indicated that the MC may be more strongly associated with elevated blood pressure, especially in patients with OSA [147,148].

Nevertheless, nowadays, special attention has been paid to the chronotherapy of hypertension, considering the circadian variations in blood pressure. People with an SBP decline during the night (non-dippers), as well as with a night-time relative SBP increase, are in a group at an elevated cardiovascular risk [149]. The recent data suggest that the EC may be associated with a higher frequency of cardiovascular events [150]. Pigazzani et al., in the Chronotype sub-study of the Treatment in Morning versus Evening (TIME) study in 2024, investigated the combined effect of the chronotype and the timing of antihypertensive drugs on cardiovascular events in subjects with arterial hypertension. The authors showed that the EC was associated with an increased risk of non-fatal myocardial infarction (MI) or stroke [151]. Thus, considering endogenous circadian rhythms by estimating the hypertensive patient’s chronotype may help to assess cardiovascular risk stratification and antihypertensive therapy.

### 6.2. Metabolic Syndrome and T2D

Metabolic syndrome (MetS) has become a significant global health problem. It is perceived as a cluster of interconnected metabolic abnormalities, which results in a significant increase in the risk of CVD and T2DM. These metabolic abnormalities usually include abdominal obesity, high blood pressure, insulin resistance, impaired glucose tolerance, hypertriglyceridemia, and low high-density lipoprotein (HDL) levels. Several criteria have been given to diagnose MetS, including the International Diabetes Foundation (IDF), the US National Cholesterol Educational Program Adult Treatment Panel III (NCEP ATP III), and the World Health Organization (WHO). The mentioned criteria may vary slightly but emphasize the same core risk factors [152,153]. What is worth mentioning is that in 2022, new criteria for MetS were presented. The novel definition of MetS involves the presence of obesity and two of the three following criteria: high blood pressure, metabolism, elevated non-high-density lipoprotein (non-HDL) cholesterol level (atherogenic dyslipidemia), and impaired glucose [154]. Observational studies indicated an association between the EC and adverse health outcomes, including metabolic disorders [155,156] and type 2 diabetes [157].

In 2018, Vera et al. published the results of a study that linked the chronotype to obesity and the development of MetS. The 2126 participants of the study were divided into two groups based on the score of the MEQ questionnaire, a 19-item scale developed by Horne and Östberg, where the cut-off point between more morning and more evening chronotype was 53 points: more MC (≥53 points; 1110 subjects) and more EC (<53 points; 1016 subjects). Both groups of patients were obese: BMI in the MC was 30.99 kg/m^2^, and in the EC, it was 31.31 kg/m^2^. The patients were measured for triglycerides, HDL, and blood pressure. The MetS score was evaluated for each subject according to the International Diabetes Federation criteria, and the HOMA-IR value was used to assess IR. The results showed that individuals representing the EC had higher blood triglyceride levels, lower HDL levels, and higher blood pressure, as well as MetS and HOMA-IR scores, compared to the MC. This suggests that obese patients with EC are more predisposed to developing metabolic disorders, Table 2 [76]. Those results were confirmed in a meta-analysis conducted by Lotti et al. The authors showed that subjects with the EC present with significantly higher glucose levels (mean difference (MD): 7.82; 95% CI: 3.18, 12.45), HbA1c (MD: 7.64; 95% CI: 3.08, 12.21), LDL cholesterol (MD: 13.69; 95% CI: 6.84, 20.54), and triglyceride concentrations (MD: 12.62; 95% CI: 0.90, 24.35) [142].

Similar conclusions were reached by the research of Vetter et al., who, in a large study, “Nurses’ Health Study 2 (NHS2)”, examined the exposure of 64 615 nurses in the years 2005–2011 to the development of T2D in the context of the chronotype and shift work. The classification of the study participants into the appropriate chronotype was based on the answer to question 19 of the MEQ: “One hears about ‘morning’ and ‘evening’ types of people. Which one of these types do you consider yourself to be?”. Based on one of five possible answers, the participants were divided into three groups: early, intermediate, and late chronotype. In each group, the BMI indicated the overweight of the respondents: early—26.4 kg/m^2^, intermediate—27.3 kg/m^2^, and late—28.6 kg/m^2^. The results collected over the 6 years of the study showed that among women who never worked shifts, those with an EC had a significantly higher risk of developing T2D than women who never worked shifts and had an MC. Additionally, it was shown that nurses with a late chronotype had a significantly increased risk of developing T2D if they did not work according to their chronotype, i.e., on day shifts that did not include night shifts, which was not observed in nurses with a late chronotype who worked night shifts. This last observation somewhat confirms the hypothesis that late-chronotype nurses who do not work according to their chronotype are more likely to develop disturbances in carbohydrate metabolism [158].

The study by Hashemipour et al. showed that glycemia control in overweight patients with already diagnosed T2D also depends on the chronotype. The 140 participants were divided into three chronotypes: MC (42 subjects), IC (40 subjects), EC (58 subjects) based on the Persian Version of MEQ. The average BMI of the participants was 27.1 kg/m^2^. The patients’ fasting glucose levels and HbA1c levels were measured, which provided information about the average blood glucose value over the past 2–3 months. The results showed that T2D patients with EC had significantly poorer glycemia control than individuals with MC (fasting glucose and HbA1c levels were 32.5 mg/dL and 1.8%, respectively, higher in the EC), Table 2 [159]. Data from the Northern Finland Birth Cohort 1966 confirmed those findings. The authors showed that the EC was associated with a higher risk of having MetS (OR 1.5; CI 95% 1.2 to 2.0) [160]. Eventually, a meta-analysis of 27 studies by Zhang et al. showed that participants with the EC had higher value of metabolic-associated parameters, such as the BMI (WMD = 0.44 kg/m^2^, 95%CI, 0.30 to 0.57 kg/m^2^, *p* < 0.001), fasting blood glucose—FBG (WMD = 5.83 mg/dL, 95%CI, 3.27 to 8.38 mg/dL, *p* < 0.001), total cholesterol level (WMD = 6.63 mg/dL, 95%CI, 0.69 to 12.56 mg/dL, *p* = 0.03), and lower HDL concentrations (WMD = −1.80 mg/dL, 95%CI, −2.30 to −1.31 mg/dL, *p* < 0.001) [161].

### 6.3. Sleep Apnea

Obstructive sleep apnea is a chronic disease defined as a recurrent pause in breathing during sleep due to partial or complete collapse of the upper respiratory airways. It results in sleep fragmentation, arousal, and oxygen desaturation at night. The total amount of individuals with OSA has been continuously growing in recent years, which highlights the importance of the disorder. Despite the impact of OSA on sleep quality, it is associated with multiple comorbidities, including cardiovascular and metabolic complications, especially among subjects struggling with obesity [162,163].

A study by Lucassen et al. in 2013 investigated the association between the chronotype and the occurrence of sleep apnea episodes in obese patients. Here, 119 participants completed the Horne and Ostberg test, which classified them into two chronotypes: 80 patients were categorized as the MC (test score: 50–86), while 39 patients were categorized as the EC (test score: 16–49). Both groups were obese, with an average BMI of 38.2 kg/m^2^ for the MC and 39.1 kg/m^2^ for the EC. Sleep parameters were monitored using wrist actigraphy monitors, and the Apnea Risk Evaluation System (Advanced Brain Monitoring Inc, Carlsbad, CA) was used to assess the number of sleep apnea episodes per hour of sleep (respiratory disturbance index—RDI), with more than five episodes per hour of sleep considered abnormal. The results showed that abnormal sleep apnea scores were nearly twice as common in the EC group than in the MC group (>5 RDI was observed in 81% of EC and only 47% of MC), Table 2 [68].

Kim et al. came to slightly different conclusions based on the Portuguese Version of MEQ results. They divided their 856 study participants into three groups: 72 patients presented an evening type (16–41 points), 338—intermediate (42–58 points), and 446—morning-type (59–86 points). The average BMI was measured for each group. In the MC group, it was 27.4 kg/m^2^; in the IC group, it was 25.9 kg/m^2^; and in the EC group, it was 26.9 kg/m^2^. The patients’ sleep quality was measured using polysomnography (PSG), and the severity of OSA was measured using the apnea-hypopnea index (AHI), which represents the number of apneas and hypopneas per hour. An AHI of 5–15/h is considered moderate, an AHI of 15–30/h is considered significant, and an AHI of >30/h is considered a severe sleep-disordered breathing disorder. In the MC group, the AHI was 8.8; in the IC group, it was 7.3; and in the EC group, it was 7.9. To further illustrate the results, the sample was stratified by BMI > 26.8 kg/m^2^ into two groups: “thin” (≤26.8 kg/m^2^), and “overweight” (>26.8 kg/m^2^) individuals. The results indicate that in the “thin” group, regardless of the chronotype, the AHI value is significantly lower than in the “overweight” group. On the other hand, in overweight individuals, the AHI is higher in those with the MC and EC than in those with the IC, Table 2 [164]. In another study, patients with OSA had a higher ME (morningness–eveningness) score, assessed using the Caen Chronotype Questionnaire, which suggests a correlation with the EC in this group [165].

Sansom et al., on the other hand, did not find any significant associations between patients’ body mass, chronotype, and the occurrence of OSA. They studied 811 patients, grouping them according to the results of the MEQ test into three chronotypes. The MC group included 326 patients with a test score of 16–52, the IC group included 347 patients with a score of 53–64, and the EC group included 138 patients with a score of 65–86. The average BMI was measured for each group. In the MC group, it was 28.2 kg/m^2^, in the IC group, it was 28.1 kg/m^2^, and in the EC group, it was 29.3 kg/m^2^. The number of sleep apnea episodes for the MC group was 13.1; for the IC group, it was 14.8, and for the EC group, it was 13.6. Based on these results, no visible trend in the relationship between the chronotype and OSA could be observed. Likewise, the adjusted mean log AHI and log T90 (sleep time with blood saturation below 90%) showed no differences between chronotypes, Table 2. However, an MC with moderate–severe OSA presented with a higher SBP than the other chronotypes, in accordance with previous studies (see division 6.1) [147].

### 6.4. Liver Diseases

The obesity pandemic is closely associated with the growing prevalence and severity of liver diseases. An excessive body weight has been linked not only to non-alcoholic fatty liver disease and simple steatosis (SS), but also with advanced diseases, such as non-alcoholic steatohepatitis (NASH), NASH-related cirrhosis, and hepatocellular carcinoma [166]. Sleep disorders have been reported to be common in subjects with NAFLD [167]. Sleeping patterns are correlated with circadian rhythms, which play an essential role in various biological functions in human metabolism. In people, the dysregulation of the circadian clock seems to be related to hepatic fat deposition and its transformation to NASH [168,169].

Maidstone et al. [170] conducted a study of 282 303 patients to investigate the association among the shift work type, chronotype, and likelihood of developing NAFLD. Based on a questionnaire (“Does your work involve shift work?”, “Does your work involve night shifts?”), the participants were divided into three groups: “day workers”, “irregular shift work”, and “permanent night-shift work”. The BMI was measured in each of these groups. For “day workers”, it was 27.08 kg/m^2^, for “irregular shift work”, it was 27.96 kg/m^2^, and for “permanent night-shift work”, it was 28.51 kg/m^2^. The chronotype was assessed in each group using a single question. Patients were questioned as to whether they consider themselves to be the following: “definitely a ‘morning’ person”, or “more a ‘morning’ than an ‘evening’ person”, “more an ‘evening’ than a ‘morning’ person”, or “definitely an ‘evening’ person”. Based on the responses, three groups were identified: MC, IC, EC. While the distribution of chronotypes among “day workers” and “irregular shift work” was similar, the number of declared ECs among “permanent night-shift work” was twice as high as among “day workers” and “irregular shift work”. The risk of NAFLD in the participants was measured using the Dallas Steatosis Index (DSI), based on the level of alanine aminotransferase, BMI, age, sex, levels of triglycerides and glucose, diabetes, hypertension, and ethnicity, ICD10 codes from hospital admissions for either NAFLD and/or NASH and the proton density fat fraction (PDFF) to describe the liver fat percentage. The results of the study clearly showed that shift workers with excessive body weight, as well as workers with a declared EC and excessive body weight, have a significantly higher risk of developing NAFLD or NASH than day workers. In line with Klawe et al. [146] and Vetter et al. [158], the authors note a higher susceptibility to obesity complications. In this case, pathological liver steatosis develops among individuals whose chronotype does not match their work hours [170,171].

Another study, conducted by Castelnuovo et al. in 2023, aimed to demonstrate the association among the chronotype, obesity, adherence to the Mediterranean diet, and the development of liver fibrosis in patients with non-alcoholic fatty liver disease (NAFLD). A total of 126 participants with an average BMI of 29.4 kg/m^2^ were studied. Based on the Munich Chronotype Questionnaire (MCTQ), 44 were classified as an MC, while 82 were classified as an IC and EC. Adherence to the Mediterranean diet was assessed using the MDS scale, with an average score of approximately 7 (out of a maximum of 14), 6.84 for the MC, and 7.09 for the IC and EC. The study found that obese patients with IC and EC chronotypes had a significantly higher risk of developing significant and advanced liver fibrosis compared to MC patients, Table 2 [172].
nutrients-17-00080-t002_Table 2Table 2Chronotype and selected metabolic complications associated with obesity.ObesityComplicationChronotype TestChronotypeBMI[kg/m^2^ ± SD]ResultsStudyHypertensionHorne–Osberg test, modified by Kwarecki, Zużewicz--A mismatch between work hours and individual chronotypes increases the risk of hypertension, ↑ BPKlawe et al. [146]Single question “What chronotype do you identify with?” 5 *25.1 ± 3.7Poor sleep pattern and EC > MC ↑ risk of developing hypertensionLv et al. [145]4 *25.4 ± 3.83 *26.0 ± 4.12 *26.9 ± 4.40 to 1 *28.0 ± 4.7Metabolicsyndrome and type 2 diabetesPersian Version of MEQMC25.9 ± 3.2EC > MC poorer glycemic control, ↑ FBG↑ HbA1cHashemipouret al. [159]IC27.3 ± 3.5EC27.8 ± 3.0.MEQMC31.4 ± 5.8EC had a significanthigher risk of T2DMMuscogiuri et al. [173]IC33.1 ± 7.3EC32.6 ± 5.5MEQMC27.0 ± 5.2MC had favorable metabolic profile WC, ↓ BMI, ↓ serum triglycerides, ↓ glucose, ↑ HDLRomanenko et al. [174]EC27.0 ± 5.7MEQMC30.9 ± 0.16EC > MC poorer glycemic control, ↑ TG, ↑ BP, ↓ HDLVera et al. [76]EC31.31 ± 0.16Single question:“What chronotype do you identify with?”early26.4 ± 5.6A mismatch between work hours and individual chronotypes increases the risk of T2DVetter et al. [158]intermediate27.3 ± 6.1late28.6 ± 6.6 MEQMC24.6 ± 2.7EC was significantly associated with diabetes (odds ratio [OR], 1.73; 95% confidence interval [CI], 1.01–2.95) and metabolic syndrome (OR, 1.74; 95% CI, 1.05–2.87)Yu et al. [155]IC24.6. ± 2.9EC24.8 ± 3.5Obstructive sleep apneaPortuguese Version of MEQMC27.4 ± 0.3IC > MC or ECSeverity of OSAKim et al. [164]IC25.9 ± 0.3EC26.9 ± 0.7MEQMC31.2 ± 6.48MC associated with aclinically meaningful increase in CPAPKnauert et al. [175]IC32.7 ± 7.19EC34.9 ± 10.4MEQMC38.2 ± 6.3EC > MC severity of OSALucassenet al. [68]EC39.1 ± 6.6MEQMC28.2 ± 5.3No relationship between chronotype and OSA; however, MC with moderate–severe OSA presented with higher SBP Sansom et al. [147]IC28.1 ± 5.3EC29.3 ± 6.8Liver diseasesMCTQMC29.05 (26.8;32.4) ***IC or EC > MCrisk of liver fibrosis with occurring NAFLDCastelnuovo et al. [172]IC + EC29.4 (25.7;32.8)Single question“What chronotype do you identify with?”“day workers” **27.08 ± 4.65Shift workers > non-shift workersEC > IC or MCrisk of developing NAFLD or NASHMaidstoneet al. [170]“irregular shift work” **27.96 ± 4.96“permanent night-shift work” **28.51 ± 4.89MEQMC45.6 ± 6.3EC was above the threshold of non-alcoholic steatohepatitis with ION index  ≥  50, higher VAI, LFE and HSIVetrani et al. [168]ICEC* healthy sleep quality score, from 0 to 5, where a higher score indicates a healthier sleep. Pattern. ** every group subdivided into MC, IC, and EC. *** median (quartile). Abbreviations: ↑— increase; ↓ — decrease; BP—blood pressure; CPAP—continuous positive airway pressure; EC—eveningness chronotype; FBG—fasting blood glucose; HbA1c—hemoglobin A1c levels; HDL—high-density lipoprotein; HSI—hepatic steatosis index; IC—intermediate chronotype; ION—index of non-alcoholic steatohepatitis; LFE—liver fat equation; MC—morningness chronotype; MCTQ—The Munich Chronotype Questionnaire; MEQ—morningness-eveningness questionnaire; NAFLD—non-alcoholic fatty liver disease; NASH—non-alcoholic steatohepatitis; OSA—obstructive sleep apnea; TG—triglycerides; T2D—type 2 diabetes; VAI—visceral adiposity index.


The cited studies indicate two general trends in the relationship between the chronotype and body weight, namely complications of overweight are more severe in individuals defined as evening chronotypes compared to those with a morning chronotype. Obesity is more severe if the individual’s work schedule (day or night shift) does not match their chronotype [146,158,170].

It is important to note some limitations when analyzing these studies [142]. Firstly, the qualification for the appropriate chronotype was based on self-completed questionnaires, which could be susceptible to conscious or unconscious falsification. Secondly, different questionnaires were used: Morningness-Eveningness Questionnaire (MEQ), Persian Version of Morning-Eventide Questionnaire, Portuguese Version of MEQ, or The Munich Chronotype Questionnaire (MCTQ). Some studies used only one question to determine the chronotype, with four or five possible answers [145,158,170]. In the Lv et al. study, the chronotype was only a component of the sleep quality score (SQC) test, which reduced the impact of the chronotype on the study results [145]. Thirdly, even in studies using the same test—the MEQ test—classification into individual chronotypes was based on different criteria—in Vera et al., classification into the EC chronotype was based on a score of 53 and above, while in Lucassen et al., it was based on a score of 50 and above and in Kim et al. a score of 59 and above [68,76,164]. Finally, not all studies directly investigated individuals who were overweight. In some studies, the average BMI of the study population indicated overweight, but not all participants were overweight [145,158]. Another limitation in the analysis of the studies mentioned above is their large heterogeneity regarding age, sex, socioeconomic status, education, ethnicity, sample size, and other factors considered, such as alcohol consumption, smoking, physical activity, type of diet and adherence to it, and sleep quality (time of going to bed, time of waking up, duration of sleep, depth, number of awakenings). Despite these limitations, the trends described in the results of the cited studies seem relatively consistent. They may set the direction for further research into the relationship between complications of overweight and the chronotype.

To summarize, there is a wealth of research showing a link between too short, too long, shallow, or interrupted sleep and the development of obesity and its metabolic or cardiovascular complications [176,177,178]. However, there is still a paucity of high-quality research directly linking the chronotype to the development of those disorders.

It is important to note that our study has some strengths and limitations. In this article, we provided a comprehensive overview to summarize recent studies on the circadian rhythm in the context of obesity. Our manuscripts bring new insight into the subject, involving the factors influencing the chronotype and the importance of physical activity and complications of obesity together. On the other hand, the main limitation of our article involves presenting data without systematic searching. Thus, it is worth mentioning that not all papers related to the subject were reviewed in this article. It also remains the goal of the future directions for the authors to describe the information mentioned above regarding the chronotype as a meta-analysis or scoping review.

## 7. Conclusions

In conclusion, the chronotype represents an individual’s preference for biological and behavioral habits dependent on the circadian rhythm. It is determined by various factors, including genetic and environmental ones, which influence i.e., meal timing and lifestyle decisions. Current evidence suggests a role of the EC in the development of obesity and obesity-related disorders. The evening chronotype has been correlated with lower sleep quality, poorer adherence to healthy eating habits, lower physical activity levels, and unfavorable metabolic changes, such as insulin resistance, higher cortisol levels, or the dysregulation of melatonin secretion. Based on the above, people with evening preferences have a higher tendency of gaining weight and a lower tendency of maintaining a healthy lifestyle. Furthermore, the complications of being overweight are less severe in the MC. People with the EC more often correspond with obesity complications, such as hypertension, disturbances in carbohydrate metabolism, metabolic syndrome, sleep apnea, or NAFLD. Since the human chronotype seems to be related to excessive body weight, patients’ health and nutritional recommendations should involve the influence of the circadian rhythm and chrononutrition. Also, following the physiological timing of energy intake may represent a novel anti-obesity approach. Nonetheless, more clinical studies with larger cohorts are needed to establish the exact relationship between the chronotype and the development of obesity and its complications.

## Figures and Tables

**Figure 1 nutrients-17-00080-f001:**
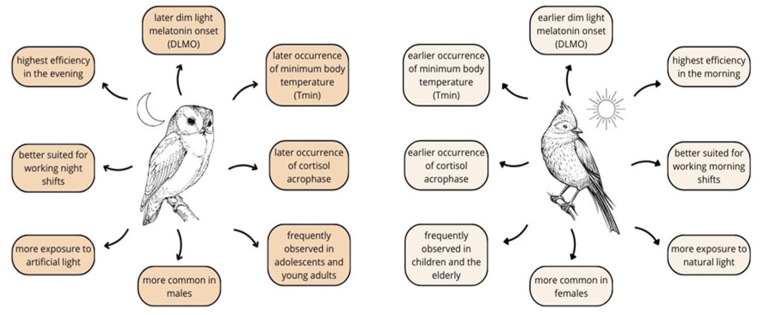
Comparison of features characterizing evening and morning chronotypes. This figure was generated using the program www.canva.com (accessed on 16 October 2024).

**Figure 2 nutrients-17-00080-f002:**
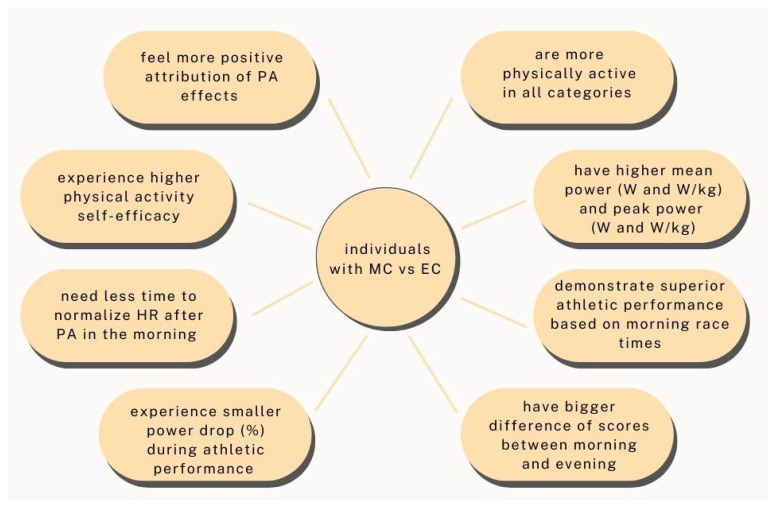
The differences in physical activity between morning and evening chronotypes. Abbreviations: EC—evening chronotype; HR—heart rate; MC—morning chronotype; PA—physical activity. This figure was made using the program www.canva.com (accessed on 24 October 2024).

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
