# Peer review of "The Role of the Chronotype in Developing an Excessive Body Weight and Its Complications—A Narrative Review"

_nutrients, 2024, doi:10.3390/nu17010080_

Round 1

Reviewer 1 Report (Previous Reviewer 2)

Comments and Suggestions for Authors

The paper us much improved in the current form. The authors replied all my request for correction. However, the title should be also change to 'narrative review' instead of overview.

Author Response

Reviewer 2 Report (Previous Reviewer 1)

Comments and Suggestions for Authors

Many thanks for the revisions made. Although the authors made some efforts to improve this manuscript, some critical concerns still remain.

1. "overview" seems quite vague. In my opinion, the scoping review might be fit with the content you aimed to present. Thus, the related reporting guideline should be followed.

Tricco, A. C., Lillie, E., Zarin, W., O'Brien, K. K., Colquhoun, H., Levac, D., ... & Straus, S. E. (2018). PRISMA extension for scoping reviews (PRISMA-ScR): checklist and explanation. Annals of internal medicine, 169(7), 467-473.

Moreover, the literature searching should be conducted in multiple available datasets to include all the related papers. Only including Medline and Pub-Med is far from satisfactory.

The keywords for searching should be carefully chosen. The full list of search strategy should also be included as a supplementary material.

2. After searching all the related papers, it is more critical about how to organize and summarize the current available findings. As far as I am concern, this is the most critical issue of the current manuscript. In the present form, it is more like a literature polymer. In most cases, each previous related study was reviewed individually. Without effective organizing the current findings in an effective way would make the readers get lost in the multiple individual findings.

3. Still, I believe the authors could try to narrow their scope and make some meta-analysis to provide specific statistical results to help the field to advance in a more scientific way.

Author Response

Reviewer 3 Report (New Reviewer)

Comments and Suggestions for Authors

Marta Pelczynska and colleagues have written a detailed review on the subject of the internal clock and body weight.

It is interesting and a very good read. I just have a few comments.

1. Please put the gene abbreviations in italics.

2. Page 2, line 54: This is not the CLOCK 3111T/C gene, but the polymorphism.

3. It is irritating that the description of the structure of the review at the end of the Introduction is written in the past tense. Present tense would be better here.

4. Some sentences are bolded. Is this a result of a previous review process? My recommendation would be not to highlight any sentences.

5. I assume that the red highlighted passages are also from a previous review process.

Round 2

Reviewer 2 Report (Previous Reviewer 1)

Comments and Suggestions for Authors

Thanks for the revisions and some concerns remain.

1. for the paragraph about the strengths and limitations, some important papers would be missing and some bias would be induced due to non-systematic searching, which should be acknowledged. It is also important to include some future directions in this part to inspire future investigations, such as meta-analysis or scoping reviews.

2. Since the authors mentioned that they did not conduct systematic searching, then it is important for the readers to get better understanding how the authors decide which papers were selected for review and the authors should provide some general search results as a supplementary material and the full search strategy would be desirably added as a supplementary material as well for the readers to judge how did you retrieve the papers and help them to judge the content expressed in this manuscript.

Author Response

This manuscript is a resubmission of an earlier submission. The following is a list of the peer review reports and author responses from that submission.

Round 1

Reviewer 1 Report

Comments and Suggestions for Authors

“The role of chronotype in developing an excessive body weight and its complications”(nutrients-3356335)

This manuscript aimed to provide a state-of-art of the topic about the role of chronotype in developing an excessive body weight and its complications. Overall, this topic is both interesting and important. The authors seem did a lot of work in drafting this manuscript. However, some concerns appeared after reading the whole manuscript.

1. Some reviews have already addressed this topic, such as Phoi, (2022), Ekiz Erim, (2023), van der Merwe (2022) mentioned in this manuscript and related reviews missing in this manuscript as listed below. Then, what the current review advance them? Why did the current manuscript still needed?

Teixeira, G. P., Guimarães, K. C., Soares, A. G. N., Marqueze, E. C., Moreno, C. R., Mota, M. C., & Crispim, C. A. (2023). Role of chronotype in dietary intake, meal timing, and obesity: a systematic review. Nutrition Reviews81(1), 75-90.

Rodríguez-Cortés, F. J., Morales-Cané, I., Rodríguez-Muñoz, P. M., Cappadona, R., De Giorgi, A., Manfredini, R., ... & López-Soto, P. J. (2022). Individual circadian preference, eating disorders and obesity in children and adolescents: a dangerous liaison? A systematic review and a meta-analysis. Children9(2), 167.

2. For each subtitle, there are also some related reviews been published, and the similar concerns appeared as in commonent1.

3. It is difficult to identity which type of review of the current manuscript belongs from the title, thus, the current title needs to be revised.

4. For table 1 and 2, how did you collect the references in both tables?It seems that the references listed are not comprehensive and some important and related papers are missing.

5. For the complications of obesity, did Hypertension, Metabolic syndrome and T2D, Sleep apnea, Liver Diseases” cover all kinds of complications of obesity?Did you mean only these complications of obesity were related to chronotype?

Reviewer 2 Report

Comments and Suggestions for Authors

In my opinion, the topic is extraordinarily important, so I am happy to read this summary. This is a huge amount of work. Despite the profound literature review, there are some controversies.

Abstract: The authors should start with a clear description of the aim of the study. It is unclear what type of review they did it and how.

Throughout the paper the reader cannot be sure what type of paper is it. Nevertheless, it sounds more than a textbook for students or a long essay subjectively summarizing the references. 

I recommend to choose first the type of review and select references according to it, see e.g., https://guides.mclibrary.duke.edu/sysreview/types 

Some of the references are outdated which should be revised.

The conclusion section is very short, lacking the becessary deepness, e.g., strength and limitations, practical implications and suggestions for future research.
